# On the Colored and the Set-Theoretical Yang–Baxter Equations

**Laszlo Barna Iantovics** [1] and **Florin Felix Nichita** [2,*]

1 George Emil Palade University of Medicine, Pharmacy, Science and Technology of Targu Mures, Gh. Marinescu 38, 540139 Tg. Mures, Romania; barna.iantovics@umfst.ro

2 Simion Stoilow Institute of Mathematics of the Romanian Academy, 010702 Bucharest, Romania

* Correspondence: florin.nichita@imar.ro; Tel.: +40-(0)-21-319-65-06; Fax: +40-(0)-21-319-65-05

**Abstract:** This paper is related to several articles published in AXIOMS, SCI, etc. The main concepts of the current paper are the colored Yang–Baxter equation and the set-theoretical Yang–Baxter equation. The Euler formula, colagebra structures, and means play an important role in our study. We show that some new solutions for a certain system of equations lead to colored Yang–Baxter operators, which are related to an Euler formula for matrices, and the set-theoretical solutions to the Yang–Baxter equation are related to means. A new coalgebra is obtained and studied.

**Keywords:** (colored) Yang–Baxter equation; set-theoretical Yang–Baxter equation; Euler's formula; classical means inequalities; coalgebra structures; artificial intelligence (AI); computational methods

**MSC:** 16W30; 16T25; 11D25; 11N45; 33B10





## 1. Introduction

After the apparition of the quantum Yang–Baxter equation in theoretical physics ([1]) and statistical mechanics ([2,3]), it also became important in quantum groups, knot theory, the quantization of integrable non-linear evolution systems, etc. (see, for example, [4–7]). However, it is "generally believed that the Yang–Baxter equation is a *fundamental mathematical structure* that will have even more relevance in future developments" (cf. [8]). The focus of the current paper is on various types of Yang–Baxter equations, and its origins are in an unpublished preprint and a series of short articles. The parameter-dependent Yang–Baxter equations are very important in mathematical physics, and, in this paper, we will adapt some constructions of solutions for them for solving for the braid condition.

In the next section, we present the colored Yang–Baxter equation. We propose the problem of finding algorithms for solving the system of Equations (5)–(9). Solutions for that system will lead to new solutions for a spectral-dependent Yang–Baxter equation. We present a new family of solutions for that system, and we hope that other solutions will be found by computational methods. Section 3 surveys recent results (see [9]) on classical means inequalities, and it presents new inequalities and interpretations. In Section 4, we explain that the new colored Yang–Baxter operator (10) is related to the Euler formula. Furthermore, by employing previous techniques, we obtain new solutions for the braid condition: the operators $R_1$, $R_2$, $R_3$, and $R_4$.

In 1992, Drinfeld formulated a number of problems in quantum group theory. He suggested considering "set-theoretical" solutions to the quantum Yang–Baxter equation. This problem attracts many beautiful minds, and the progress in solving it is significant (see [10–14]). Reference [12] studies the Yang–Baxter and pentagon equations applicable in mathematical physics. The objective of this research consists of studying how the solutions of the pentagon equation are a useful approach for obtaining new solutions of the Yang–Baxter equation. We will recall the terminology and some results for the set-theoretical Yang–Baxter equation. One can interpret the set-theoretical Yang–Baxter equation as a condition that unifies the equivalence relations and the order relations. Means are also related to the set-theoretical Yang–Baxter equation.

Some of the above ideas lead to other new constructions related to Euler's formula. In Section 6, this investigation leads to a new coalgebra structure. Some of its properties are highlighted. A short discussion on Artificial Intelligence and computational methods concludes our paper.

## 2. The Colored Yang–Baxter Equation

We will work over a field $k$, and the tensor products are defined over $k$. For $V$, a $k$-space, we denote by $\tau : V \otimes V \to V \otimes V$ the twist map defined by $\tau(v \otimes w) = w \otimes v$, and by $I : V \to V$ the identity map of the space $V$. The following terminology will be used, when we refer to various versions of Yang–Baxter equations. If $R : V \otimes V \to V \otimes V$ is a $k$-linear map, then $R^{12} = R \otimes I, R^{23} = I \otimes R, R^{13} = (I \otimes \tau)(R \otimes I)(I \otimes \tau)$.

**Definition 1.** *A hypothesis that the k-linear map $R : V \otimes V \to V \otimes V$ is invertible is usually required.*

Now, $R$ is a Yang–Baxter operator if it satisfies the equation

$$R^{12} \circ R^{23} \circ R^{12} = R^{23} \circ R^{12} \circ R^{23} \tag{1}$$

**Remark 1.** *To avoid confusion, we call Equation (1) the braid equation. The operator R satisfies (1) if and only if $R \circ \tau$ satisfies the constant quantum Yang–Baxter equation (QYBE), if and only if $\tau \circ R$ satisfies the constant QYBE:*

$$R^{12} \circ R^{13} \circ R^{23} = R^{23} \circ R^{13} \circ R^{12} \tag{2}$$

**Remark 2.**    *(i) The simplest examples of Yang–Baxter operators are $\tau : V \otimes V \to V \otimes V$ and its slight generalizations.*

   *(ii) Using computational methods, the invertible solutions for (2) in Dimension 2 were classified in [15].*

   *(iii) For dimensions greater than 2, the problem of classifying the Yang–Baxter operators is an unsolved problem.*

We now consider an associative algebra $A$ over $k$, $\alpha, \beta, \gamma \in k$, and the $k$-linear map: $R_{\alpha,\beta,\gamma}^A : A \otimes A \to A \otimes A$, $R_{\alpha,\beta,\gamma}^A(a \otimes b) = \alpha ab \otimes 1 + \beta 1 \otimes ab - \gamma a \otimes b$.

**Theorem 1** (Dăscălescu and Nichita [4]). *With the above notation, if $\dim A \geq 2$, then $R_{\alpha,\beta,\gamma}^A$ is a Yang–Baxter operator if and only if one of the following holds: (i) $\alpha = \gamma \neq 0$, $\beta \neq 0$; (ii) $\beta = \gamma \neq 0$, $\alpha \neq 0$; (iii) $\alpha = \beta = 0$, $\gamma \neq 0$.*
*Notice that $(R_{\alpha,\beta,\gamma}^A)^{-1} = R_{\frac{1}{\beta},\frac{1}{\alpha},\frac{1}{\gamma}}^A$ in Cases (i) and (ii) (and $(R_{0,0,\gamma}^A)^{-1} = R_{0,0,\frac{1}{\gamma}}^A$).*

There are many versions of the Yang–Baxter equation. We will consider the unification of a one-parameter-dependent Yang–Baxter equation and a two-parameter-dependent Yang–Baxter equation, which is sometimes called the "colored Yang–Baxter equation" (see, also, [6]).

We define a colored Yang–Baxter operator as a function

$$R : k \times X \times X \to End_k V \otimes V,$$

where $V$ is a finite dimensional $k$-space, and $X$ is a set.

So, if we fix $x \in k$, $u, v \in X$, $R(x, u, v) : V \otimes V \to V \otimes V$ is a linear operator.

According to our previous terminology, there exist three operators acting on a triple tensor product $V \otimes V \otimes V$: $R^{12}(x, u, v) = R(x, u, v) \otimes I$, $R^{23}(x, v, w) = I \otimes R(x, v, w)$, and $R^{13}(x, u, w)$ (an operator acting non-trivially on the first and third factors in $V \otimes V \otimes V$).

*R* is a colored Yang–Baxter operator if it satisfies the equation:

$$R^{12}(x,u,v)R^{13}(x+y,u,w)R^{23}(y,v,w) = R^{23}(y,v,w)R^{13}(x+y,u,w)R^{12}(x,u,v) \quad (3)$$

for all $x, y \in k$, $u, v, w \in X$.

We want to apply our method for finding solutions for Equation (3).

Let *X* be (a subset of) the ground field *k*, and $V = A$, an associative *k*-algebra. We will construct a solution to Equation (3) from the associativity of the product in *A*. So, we are looking for solutions to Equation (3) of the following form

$$R(x,u,v)(a \otimes b) = \alpha_x(u,v)1 \otimes ab + \beta_x(u,v)ab \otimes 1 - \gamma_x(u,v)b \otimes a, \quad (4)$$

where $\alpha_x, \beta_x, \gamma_x$ are *k*-valued functions on $X \times X$ for any $x \in k$.

Inserting this ansatz into Equation (3), we obtained the following system of equations (whose solutions produce colored Yang–Baxter operators):

$$\begin{aligned}(\beta_y(v,w) - \gamma_y(v,w))&(\alpha_x(u,v)\beta_{x+y}(u,w) - \alpha_{x+y}(u,w)\beta_x(u,v))\\ +(\alpha_x(u,v) - \gamma_x(u,v))&(\alpha_y(v,w)\beta_{x+y}(u,w) - \alpha_{x+y}(u,w)\beta_y(v,w)) = 0\end{aligned} \quad (5)$$

$$\begin{aligned}\beta_y(v,w)(\beta_x(u,v) - \gamma_x(u,v))&(\alpha_{x+y}(u,w) - \gamma_{x+y}(u,w))\\ +(\alpha_y(v,w) - \gamma_y(v,w))&(\beta_{x+y}(u,w)\gamma_x(u,v) - \beta_x(u,v)\gamma_{x+y}(u,w)) = 0\end{aligned} \quad (6)$$

$$\begin{aligned}\alpha_x(u,v)\beta_y(v,w)(\alpha_{x+y}(u,w) - \gamma_{x+y}(u,w)) &+ \alpha_y(v,w)\gamma_{x+y}(u,w)(\gamma_x(u,v) - \alpha_x(u,v))\\ +\gamma_y(v,w)(\alpha_x(u,v)\gamma_{x+y}(u,w) &- \alpha_{x+y}(u,w)\gamma_x(u,v)) = 0\end{aligned} \quad (7)$$

$$\begin{aligned}\alpha_x(u,v)\beta_y(v,w)(\beta_{x+y}(u,w) - \gamma_{x+y}(u,w)) &+ \beta_y(v,w)\gamma_{x+y}(u,w)(\gamma_x(u,v) - \beta_x(u,v))\\ +\gamma_y(v,w)(\beta_x(u,v)\gamma_{x+y}(u,w) &- \beta_{x+y}(u,w)\gamma_x(u,v)) = 0\end{aligned} \quad (8)$$

$$\begin{aligned}\alpha_x(u,v)(\alpha_y(v,w) - \gamma_y(v,w))&(\beta_{x+y}(u,w) - \gamma_{x+y}(u,w))\\ +(\beta_x(u,v) - \gamma_x(u,v))&(\alpha_{x+y}(u,w)\gamma_y(v,w) - \alpha_y(v,w)\gamma_{x+y}(u,w)) = 0\end{aligned} \quad (9)$$

**Remark 3.** *(i) If we consider the system of Equations (5)–(9), one simplification is to require that $\alpha_x, \beta_x$, and $\gamma_x$ do not depend on x, u, and v. So, we denote them $\alpha, \beta$, and $\gamma$. Letting $\gamma$ be an arbitrary constant, and $a \in k$, one verifies that setting $\alpha = a\gamma$ and $\beta = \gamma$ gives a solution. Similarly, letting $\gamma$, $b \in k$ be arbitrary, setting $\alpha = \gamma$ and $\beta = b\gamma$ gives a solution.*

*(ii) The following is a new family of solutions for the system of Equations (5)–(9):*

$$\alpha_x(u,v) = p(a^x u - v), \quad \beta_x(u,v) = q(a^x u - v), \quad \text{and} \quad \gamma_x(u,v) = pa^x u - qv, \text{ where}$$
*$a, p, q \in k$.*

*Here, a is an arbitrary element of the field k. We do need that k is closed under exponentiation, but this is true for **R** and **C**, which are the fields we are mainly interested in.*

We can organize the results from this section as a theorem.

**Theorem 2.** *Under the above assumptions, there exists the following family of solutions for Equation (3):*

$$R(x,u,v)(c \otimes d) = p(a^x u - v)1 \otimes cd + q(a^x u - v)cd \otimes 1 - (pa^x u - qv)d \otimes c. \quad (10)$$

**Proof.** We will only check that $\alpha_x(u,v) = p(a^x u - v)$, $\beta_x(u,v) = q(a^x u - v)$, and $\gamma_x(u,v) = pa^x u - qv$ are solutions for Equation (5). However, $\alpha_x(u,v)\beta_{x+y}(u,w) = \alpha_{x+y}(u,w)\beta_x(u,v)$ and $\alpha_y(v,w)\beta_{x+y}(u,w) = \alpha_{x+y}(u,w)\beta_y(v,w)$. $\square$

### 3. Unification Results

We will review and enhance a collection of recent results (see [9]) on unification theories in order to relate them to the previous section.

The harmonic mean: $H = H(a_1, \ldots, a_n) = \frac{n}{\frac{1}{a_1} + \cdots + \frac{1}{a_n}}$, $a_1, \ldots, a_n \in \mathbb{R}^*$;

The geometric mean: $G = G(a_1, \ldots, a_n) = \sqrt[n]{a_1 a_2 \ldots a_n}$, $a_1, \ldots, a_n \in \mathbb{R}_+$;

The arithmetic mean: $A = A(a_1, \ldots, a_n) = \frac{a_1 + \cdots + a_n}{n}$, $a_1, \ldots, a_n \in \mathbb{R}$;

The quadratic mean: $Q = Q(a_1, \ldots, a_n) = \sqrt{\frac{a_1^2 + \cdots + a_n^2}{n}}$, $a_1, \ldots, a_n \in \mathbb{R}$;

The projection type i-mean: $P_i^m = P_i^m(a_1, \ldots, a_n) = \frac{1}{m+n-1}a_1 + \cdots + \frac{m}{m+n-1}a_i + \cdots + \frac{1}{m+n-1}a_n$, $a_1, \ldots, a_n \in \mathbb{R}$.

The generalized mean: $m(\alpha) = m_\alpha(a_1, \ldots, a_n) = \sqrt[\alpha]{\frac{a_1^\alpha + \cdots + a_n^\alpha}{n}}$, $a_1, \ldots, a_n \in \mathbb{R}_+$, $\alpha \geq 1$, $\alpha \in \mathbb{N}$.

**Remark 4.** *We note that $m(1) = A$ and $m(2) = Q$ for $a_1, \ldots, a_n \in \mathbb{R}_+$. Let us consider the function $m(\alpha) = \left(\frac{a_1^\alpha + \cdots + a_n^\alpha}{n}\right)^{\frac{1}{\alpha}}$, $\alpha \in \mathbb{R}^*$. One can extend the function $m(\alpha)$ to $\mathbb{R}$ by assigning $m(0) = m_G$. This is a natural extension, as $\lim_{\alpha \to 0} N_\alpha(a_1, \ldots, a_n) = m_G(a_1, \ldots, a_n)$. It makes sense now to write: $m(0) = G$ and $m(-1) = H$.*

**Remark 5.** *Because $\lim_{\alpha \to \infty} m(\alpha) = Max\{a_1, \ldots, a_n\}$, $\lim_{\alpha \to -\infty} m(\alpha) = Min\{a_1, \ldots, a_n\}$, and $\lim_{m \to \infty} P_i^m = a_i$, we will regard $Max\{a_1, \ldots, a_n\}$, $Min\{a_1, \ldots, a_n\}$ and $P_i(a_1, \ldots, a_n) = a_i$ as means.*

We consider the following generalization of the above function $m(\alpha)$.

**Theorem 3.** *Let $O : \mathbb{R} \times \mathbb{R} \to \mathbb{R}$, be defined by the following rule:*

$$O(x, y) = \begin{cases} \left(\frac{a_1^x + \cdots + a_n^x}{a_1^y + \cdots + a_n^y}\right)^{\frac{1}{x-y}}, & \text{if } x \neq y; \\ e^{\frac{a_1^x \ln a_1 + \cdots + a_n^x \ln a_n}{a_1^x + \cdots + a_n^x}} = a_1^{\frac{a_1^x}{a_1^x + \cdots + a_n^x}} \ldots a_n^{\frac{a_n^x}{a_1^x + \cdots + a_n^x}}, & \text{otherwise.} \end{cases}$$

*Then, the function M has the following properties:*

1. *It is a continuous function.*
2. *$O(x, y) = O(y, x) \quad \forall\, x, y \in \mathbb{R}$.*
3. *For any fixed $x_0 \in \mathbb{R}$, the function $f(y) = O(x_0, y)$ is a strongly increasing function.*
4. *For any fixed $y_0 \in \mathbb{R}$, the function $g(x) = O(x, y_0)$ is a strongly increasing function.*

**Remark 6.** *We can easily see that $O(-1, 0) = H$, $O(0, 0) = G$, $O(1, 0) = A$, $O(2, 0) = Q$, and, in general, $O(\alpha, 0) = m(\alpha)$.*

The following are examples of inequalities from [9], for $a > 0$, $b > 0$:

$$\sqrt{ab} \leq \left(\frac{\sqrt{a} + \sqrt{b}}{2}\right)^2 \leq \frac{a+b}{2} \leq \left(\frac{a^{\frac{3}{2}} + b^{\frac{3}{2}}}{2}\right)^{\frac{2}{3}} \leq \sqrt{\frac{a^2 + b^2}{2}} \leq a + b - \sqrt{ab}. \tag{11}$$

Of course, one can translate the above inequalities in terms of relations of the above function $O(x, y)$.

The next inequalities are new:

**Theorem 4.** *The following inequalities hold for positive real numbers:*

$$\log_2(2^{\sqrt{a}} + 2^{\sqrt{b}} - 1) \le \sqrt{ab} \le \frac{a+b}{2} \le \log_{\sqrt{2}}(\sqrt{2}^a + \sqrt{2}^b) - 2 .$$

**Proof.** The first inequality follows from algebraic manipulations: $2^{\sqrt{a}} + 2^{\sqrt{b}} - 1 \le 2^{\sqrt{ab}} \iff (2^{\sqrt{a}} - 1)(2^{\sqrt{b}} - 1) \ge 0$. The last inequality follows from the convexity of the function $f(x) = \sqrt{2}^x$: $\frac{(\sqrt{2}^a + \sqrt{2}^b)}{2} \ge \sqrt{2}^{\frac{a+b}{2}}$. $\square$

**Remark 7.** *We consider new operations on* $\mathbb{R}$: $a \circ b = 2^{\sqrt{a}} + 2^{\sqrt{b}} - 1$, $a * b = \log_{\sqrt{2}}(\sqrt{2}^a + \sqrt{2}^b)$.

*The first operation is some kind of exponentiation (in particular, $a \circ a = 2^{\sqrt{a}+1} - 1$).*
*The last operation "generates" the addition: $a * a = a + 2$, $a * a * a * a = a + 4$.*
*The above theorem can be stated as:*

$$\log_2(a \circ b) \le \sqrt{ab} \le \frac{a+b}{2} \le a * b - 2 .$$

**Remark 8.** *It is an open problem to generalize the above inequalities. Possible generalizations might be the following inequalities:* $\log_3(3^{\sqrt[3]{ab}} + 3^{\sqrt[3]{bc}} + 3^{\sqrt[3]{ca}} - 3^{\sqrt[3]{a}} - 3^{\sqrt[3]{b}} - 3^{\sqrt[3]{c}} + 1) \le \sqrt[3]{abc} \le \frac{a+b+c}{3} \le \log_{\sqrt[3]{3}}(\sqrt[3]{3}^a + \sqrt[3]{3}^b + \sqrt[3]{3}^c) - 3$, $\forall a, b, c > 0$.

### 4. Generalized Euler Formula

It is useful to identify linear applications with matrices in the current section. Additionally, the exponential function of a matrix is thought as the Taylor expansion of the real exponential function evaluated at the given matrix. The ring of all $n \times n$-matrices over the field $k$ is denoted by $M_n(k)$. $I$ will be the identity matrix in $M_4(k)$, and $I'$ will be the identity matrix in $M_2(k)$.

**Theorem 5 ([9]).** *Let $J \in M_n(\mathbb{C})$ such that*

$$(J \otimes I') \circ (I' \otimes J) = (I' \otimes J) \circ (J \otimes I') \tag{12}$$

*If $e^{zJ} = R(z)$, $z \in \mathbb{C}$, then the following spectral-dependent Yang–Baxter equation is satisfied:*

$$(R \otimes I')(z) \circ (I' \otimes R)(z+w) \circ (R \otimes I')(w) = (I' \otimes R)(w) \circ (R \otimes I')(z+w) \circ (I' \otimes R)(z) . \tag{13}$$

**Theorem 6 ([9]).** *For $J \in M_n(\mathbb{C})$, $J^2 \in \mathbb{C}I$, there exist two "pseudo-trigonometric" functions $c, s : \mathbb{C} \to \mathbb{C}$ such that:*

$$e^{zJ} = c(z)I + s(z)J . \tag{14}$$

*Moreover, the functions $c$ and $s$ have the following properties:*

(i) *If $J^2 = \alpha I$, then $c(z + w) = c(z)c(w) + \alpha s(z)s(w)$, $s(z + w) = s(z)c(w) + c(z)s(w)$;*
(ii) *If $\alpha = 0$ in the above case, then $c(z) = 1$ and $s(z) = z$, or $c(z) = 0$ and $s(z) = 0$;*
(iii) *If $J^2 = \beta^2 I$, with $\beta^2 = \alpha \ne 0$, then $c(z) = \cosh(\beta z)$ and $s(z) = \frac{\sinh(\beta z)}{\beta}$.*

**Remark 9.** *We now present examples of how the above theorem can be applied, and we also return to the braid condition (1) in order to obtain new solutions of it in the cases where this is possible.*

(i) *If $J = i\,I$, Formula (14) is equivalent to $e^{ix} = \cos x + i \sin x$.*

*If we consider the complex evaluated matrix ($c, d \in \mathbb{C}$)* $J = \begin{pmatrix} 0 & 0 & c & d \\ 0 & 0 & 0 & c \\ 0 & 0 & 0 & 0 \\ 0 & 0 & 0 & 0 \end{pmatrix}$

then, $J^2 = 0_4$, $J^{12}J^{23} = J^{23}J^{12}$, $I + Jx = e^{xJ} = R(x)$, and

$$(R \otimes I')(x) \circ (I' \otimes R)(x+y) \circ (R \otimes I')(y) = (I' \otimes R)(y) \circ (R \otimes I')(x+y) \circ (I' \otimes R)(x).$$

(ii) If $J = \begin{pmatrix} 0 & 0 & 0 & 1 \\ 0 & 0 & i & 0 \\ 0 & i & 0 & 0 \\ -1 & 0 & 0 & 0 \end{pmatrix}$ then $e^{zJ} = \cos(z)I + \sin(z)J$, which is a solution for (13).

In this case, two interesting solutions for the braid condition (1) can be obtained.

We consider the operator $R = aI + bJ$, with $a, b \in \mathbb{C}$ and $J$ the above matrix. The braid condition is satisfied by $R$ if $a^2 = -b^2$. The following two main cases are of interest:

$$R_1 = \begin{pmatrix} i & 0 & 0 & 1 \\ 0 & i & i & 0 \\ 0 & i & i & 0 \\ -1 & 0 & 0 & i \end{pmatrix};$$

$$R_2 = \begin{pmatrix} -i & 0 & 0 & 1 \\ 0 & -i & i & 0 \\ 0 & i & -i & 0 \\ -1 & 0 & 0 & -i \end{pmatrix}.$$

(iii) If $J = \begin{pmatrix} 0 & 0 & 0 & 1 \\ 0 & 0 & 1 & 0 \\ 0 & 1 & 0 & 0 \\ 1 & 0 & 0 & 0 \end{pmatrix}$ then $e^{zJ} = \cosh(z)I + \sinh(z)J$ ; this is also a solution for (13).

In this case, two other interesting solutions for the braid condition (1), can be obtained. In a similar way with Case (ii), we consider the operator $R = aI + bJ$, with $a, b \in \mathbb{C}$ and $J$ our matrix. The braid condition is satisfied by $R$ if $a^2 = b^2$. The following two main cases are of interest:

$$R_3 = \begin{pmatrix} 1 & 0 & 0 & 1 \\ 0 & 1 & 1 & 0 \\ 0 & 1 & 1 & 0 \\ 1 & 0 & 0 & 1 \end{pmatrix};$$

$$R_4 = \begin{pmatrix} 1 & 0 & 0 & -1 \\ 0 & 1 & -1 & 0 \\ 0 & -1 & 1 & 0 \\ -1 & 0 & 0 & 1 \end{pmatrix}.$$

**Remark 10.** *Operator (10) has the form (14).*

*More precisely, we start with* $R(x, u, v)(c \otimes d) = p(a^x u - v)1 \otimes cd + q(a^x u - v)cd \otimes 1 - (pa^x u - qv)d \otimes c$ *and consider* $R'(x, u, v) = -\tau \circ R(x, u, v)$.

*We take* $u = v$, $a = e$, $p = 1$, *and* $q = -1$. *After simplifications, we obtain an operator of the form* $R''(x) = I + Jx = e^{xJ}$.

*If we choose the algebra* $\frac{k[X]}{X^2 - aX - b}$, *we can express* $R''(x)$ *as a matrix:*

$$R''(x) = \begin{pmatrix} 1 & 0 & 0 & 0 \\ 0 & x+1 & x & ax \\ 0 & -x & 1-x & -ax \\ 0 & 0 & 0 & 1 \end{pmatrix}$$

## 5. The Set-Theoretical Yang–Baxter Equation

If $X$ is a set, let $S : X \times X \to X \times X$ be a function, $S^{12} = S \times I$ and $S^{23} = I \times S$.

**Definition 2.** *Using the above notation, the set-theoretical Yang–Baxter equation reads:*

$$S^{12} \circ S^{23} \circ S^{12} = S^{23} \circ S^{12} \circ S^{23} \tag{15}$$

**Definition 3.** *We are using the following notation for a relation $R$ on the set $X$: we denote by $R^{op}$ the opposite relation of $R$, and we denote by $\bar{R}$ the complementary relation of $R$. Let $\Delta = \Delta_X = \{(x, x) \mid \forall x \in X\} \subset X \times X$.*

**Theorem 7** (Hobby and Nichita [16]). *We consider a reflexive relation $R \subset X \times X$ on the set $X$. If we define the function $S = S_R : X \times X \to X \times X$ by $S(u, v) = \begin{cases} (u, v), & \text{if } (u, v) \in R \\ (v, u); & \text{otherwise,} \end{cases}$ then, $S$ satisfies (15) if and only if $R \cup R^{op}$ is an equivalence relation, and $\bar{R}$ is a strict partial order relation on each class of $R \cup R^{op}$.*

Note that the definition of $S_R$ in the above theorem makes sense for any relation $R$, and that $S_R$ only depends on $R \cap \bar{\Delta}$.

**Remark 11.** *The following properties hold.*

(i) *For any relation $R$, $S_R = S_{R \cup \Delta}$.*
(ii) *If $R$ is a symmetric relation, then $S_R = S_{R^{op}}$.*
(iii) *If $R$ is an antisymmetric relation, then $T \circ S_R = S_{R^{op}}$, $[T : X \times X \to X \times X, (x, y) \mapsto (y, x)]$.*
(iv) *For any relation $R$, $T \circ S_R = S_{\bar{R}}$.*
(v) *For any equivalence relation $R$, $S_R^{12} \circ S_R^{23} \circ S_R^{12} = S_R^{23} \circ S_R^{12} \circ S_R^{23}$.*
(vi) *For any partial order relation $R$, $S_R^{12} \circ S_R^{23} \circ S_R^{12} = S_R^{23} \circ S_R^{12} \circ S_R^{23}$.*
(vii) *For any total order relation $R$, $S_R(u, v) = (Max(u, v), Min(u, v))$.*

**Theorem 8.** *For $z, w \in \mathbb{C}$, such that $z = \rho e^{i\alpha}$, $w = \rho' e^{i\beta}$, with $\alpha, \beta \in [0, 2\pi)$, we define:*

$$Q(z, w) = \frac{z + iw}{\sqrt{2}}, \quad A(z, w) = \frac{z + w}{2}, \quad G(z, w) = \sqrt{\rho \rho'} e^{i\frac{\alpha + \beta}{2}}, \quad H(z, w) = \begin{cases} \frac{2zw}{z+w}, & \text{if } z \neq -w \\ 0, & \text{otherwise} \end{cases} \tag{16}$$

*If $\alpha = \beta$ or $\rho_{Max} \geq 7\rho_{min}$, then:*

$$\|H\| \leq \|G\| \leq \|A\| \tag{17}$$

**Sketch of proof.** If $\alpha = \beta$, the inequalities (17) are equivalent to the classical means inequalities; moreover, we have $\|H\| \leq \|G\| \leq \|A\| \leq \|Q\|$.

The inequality $\|G\| \leq \|A\|$ is equivalent to finding $x \in \mathbb{R}$ such that $x^2 + 2[\cos(\alpha - \beta) - 2]x + 1 \geq 0$.

The last part of the theorem follows from the property $\|HA\| = \|G^2\|$. □

**Theorem 9.** *The map $R : \mathbb{C} \times \mathbb{C} \to \mathbb{C} \times \mathbb{C}$, $R(z, w) = (A(z, w), z)$ is a solution for (15).*
*The map $R : \mathbb{C} \times \mathbb{C} \to \mathbb{C} \times \mathbb{C}$, $R(z, w) = (G(z, w), z)$ is a solution for (15).*

**Proof.** We see that

$$R^{12} \circ R^{23} \circ R^{12}(z, w, u) = R^{12} \circ R^{23}(\frac{z + w}{2}, z, u) = R^{12}(\frac{z + w}{2}, \frac{z + u}{2}, z) = (\frac{\frac{z+w}{2} + \frac{z+u}{2}}{2}, \frac{z + w}{2}, z).$$

In a similar manner, $R^{23} \circ R^{12} \circ R^{23}(z, w, u) = (\frac{z + \frac{w+u}{2}}{2}, \frac{z+w}{2}, z)$.

The last part of the theorem follows below.

$$R^{12} \circ R^{23} \circ R^{12}(z, \quad w, \quad u) \quad = \quad R^{12} \circ R^{23}(\sqrt{\rho \rho'} \; e^{i\frac{\alpha+\beta}{2}}, \quad z, \quad u) \quad =$$
$$R^{12}(\sqrt{\rho \rho'} \; e^{i\frac{\alpha+\beta}{2}}, \; \sqrt{\rho \rho''} \; e^{i\frac{\alpha+\gamma}{2}}, \; z) = (\sqrt{\rho \sqrt{\rho' \rho''}} \; e^{i\frac{2\alpha+\beta+\gamma}{4}}, \; \sqrt{\rho \rho'} \; e^{i\frac{\alpha+\beta}{2}}, \; z).$$

Additionally, $R^{23} \circ R^{12} \circ R^{23}(z, w, u) = (\sqrt{\rho \sqrt{\rho' \rho''}} \; e^{i\frac{\alpha+\frac{\beta+\gamma}{2}}{2}}, \; \sqrt{\rho \rho'} \; e^{i\frac{\alpha+\beta}{2}}, \; z).$

□

**Remark 12.** *Let us restrict our attention to positive real numbers. Rephrasing the above theorem, we can say that $R = A \times P_1$ and $R = G \times P_1$ are solutions for (15). Additionally, $R = Max \times Min$ is a solution for (15). According to Remark 11 (vii), this solution is related to Theorem 7.*

## 6. Coalgebra Structures

Let us start by observing that Formula (14) can be interpreted in terms of coalgebras (see [17,18]).

There exists a coalgebra $\frac{\mathbb{C}[X]}{X}^2 = \mathbb{C}[a]$, where $a^2 = \alpha = \beta^2 \in \mathbb{C}$, generated by two generators $c$ and $s$, such that: $\Delta(c) = u \otimes u + \alpha s \otimes s$, $\Delta(s) = c \otimes s + s \otimes c$, $\varepsilon(c) = 1$, $\varepsilon(s) = 0$.

Formula (14) leads to the subcoalgebra generated by $u + \beta s$.

It is an open problem how to interpret Equation (13) in the above representative coalgebras.

**Theorem 10.** *There exists a coalgebra structure over k, generated by three generators a, b, and c, such that: $\Delta(a) = a \otimes c + b \otimes b + c \otimes a$, $\Delta(b) = a \otimes a + b \otimes c + c \otimes b$, $\Delta(c) = a \otimes b + b \otimes a + c \otimes c$, $\varepsilon(a) = 0 = \varepsilon(b)$, $\varepsilon(c) = 1$.*

**Proof.** There exists a direct proof.

Alternatively, let $A \in M_2(k)$ such that $A^3 = I$. We are looking for functions $\alpha_x$, $\beta_x$, $\gamma_x$ such that $(\alpha_x A + \beta_x A^2 + \gamma_x I)(\alpha_y A + \beta_y A^2 + \gamma_y I) = (\alpha_{x+y} A + \beta_{x+y} A^2 + \gamma_{x+y} I)$, and $\alpha_0 = 0$, $\beta_0 = 0$, $\gamma_0 = 1$. The system of equations

$$\alpha_{x+y} = \alpha_x \gamma_y + \beta_x \beta_y + \gamma_x \alpha_y,$$

$$\beta_{x+y} = \alpha_x \alpha_y + \beta_x \gamma_y + \gamma_x \beta_y,$$

$$\gamma_{x+y} = \alpha_x \beta_y + \beta_x \alpha_y + \gamma_x \gamma_y,$$

leads to a representative coalgebra. □

**Remark 13.** *If the coalgebra structure from the previous theorem is defined over $M_2(k)$, and the matrix $A \in M_2(k)$ has the property $A^3 = I$, then there exists a coideal generated by $Aa + A^2b + c$. Indeed, one can check that $\Delta(Aa + A^2b + c) = (Aa + A^2b + c) \otimes (Aa + A^2b + c)$.*

*This is true:*

$$\Delta(Aa + A^2b + c) = Aa \otimes c + Ab \otimes b + Ac \otimes a + A^2a \otimes a + A^2b \otimes c + A^2c \otimes b + a \otimes b + b \otimes a + c \otimes c;$$

$$(Aa + A^2b + c) \otimes (Aa + A^2b + c) = A^2a \otimes a + a \otimes b + Aa \otimes c + b \otimes a + Ab \otimes b + A^2b \otimes c + Ac \otimes a + A^2c \otimes b + c \otimes c.$$

## 7. Final Comments and Conclusions

Computational methods are important tools in some areas of abstract mathematics and for certain teaching methods. In our current paper, we referred to the computational methods used for finding all invertible solutions for the quantum Yang–Baxter equation (Equation (2)) in the case $n = 2$ (see [15,19]). These algorithms are not powerful enough to fully classify the solutions for other small dimensions. Thus, a complete computer calculation for $n = 3$ is still out of reach at this time. Additionally, one can explicitly solve the set-theoretical Yang–Baxter equation (Equation (15)) for small sets by an exhaustive search. So, computational methods are helpful for solving (other) mathematical equations (see [20–23], or the system of Equations (5)–(9)).

This paper is related to several articles published in AXIOMS and SCI, where examples of unification constructions in mathematics are presented, or poetical approaches are proposed. Unifying mathematical structures is not always an easy task, and this guiding principle has subtle rules. So, there exists a universal unifying principle both in mathematics and in poetry.

Some of our Yang–Baxter operators are related to quantum gates. Quantum computers are gaining more interest nowadays, as they are very powerful. Artificial Intelligence as we understand it today, was born during the summer of 1956 (at Dartmouth College). The Romanian history of informatics (see [24]) could also be traced back around that period. Thanks to a summer grant at the Targu Mures Computer Center in 1987 (for studying informatics on HC 85, using *Basic* language), we implemented that technology in education.

**Author Contributions:** The curent paper resulted from a long term collaboration, meetings at conferences and some short communications. Conceptualization, F.F.N.; supervision, L.B.I.; writing—original draft preparation, F.F.N.; resources, L.B.I.; writing—review and editing, L.B.I. and F.F.N. All authors have read and agreed to the published version of the manuscript.

**Funding:** This research received no external funding.

**Acknowledgments:** We would like to thank David Hobby (SUNY New Paltz).

**Conflicts of Interest:** The authors declare no conflict of interest.

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
