# Peer review of "On the Colored and the Set-Theoretical Yang–Baxter Equations"

_axioms, doi:10.3390/axioms10030146_

Round 1
Reviewer 1 Report
This paper is devoted to the colored generalizaton of the Yang-Baxter formula. It uses several new concepts and derives new solutions. It is well written, and contains a lot of novel material. I recommend it for publication in "Axioms"
Author Response
Dear Reviewer,
Thank you for your report.
Best wishes !
Reviewer 2 Report
Remark 2.4 \alpha, \beta, \gamma do not depend on x.
p.6 Taylor expansion
p.9 Better sketch also last part and remove "The last part of the theorem is left for the reader."
I suggest that the conclusions should be rewritten:
The first sentence is obvious. What would be important is to explain conceptually the links between the several articles mentioned. This should be the main part of the conclusions.
AI is not mentioned in the text except in the conclusions. This comes a bit as a surprise. This part of the text should be integrated in such a way that it is not perceived as an unconnected addendum.
Author Response
Dear Reviewer,
We have made the corrections.
Thank you !
Reviewer 3 Report
The manuscript is written in the clear manner and it brings interesting results.
Author Response
Dear Reviewer,
Thank you very much for your report.
Best wishes !